# Mesenchymal Stem Cells and MSCs-Derived Extracellular Vesicles in Infectious Diseases: From Basic Research to Clinical Practice

**DOI:** 10.3390/bioengineering9110662

**Published:** 2022-11-08

**Authors:** Natalia Yudintceva, Natalia Mikhailova, Viacheslav Fedorov, Konstantin Samochernych, Tatiana Vinogradova, Alexandr Muraviov, Maxim Shevtsov

**Affiliations:** 1Institute of Cytology of the Russian Academy of Sciences (RAS), St. Petersburg 194064, Russia; 2Personalized Medicine Centre, Almazov National Medical Research Centre, St. Petersburg 197341, Russia; 3Saint-Petersburg State Research Institute of Phthisiopulmonology of the Ministry of Health of the Russian Federation, St. Petersburg 191036, Russia

**Keywords:** mesenchymal stem cells, extracellular vesicles, regenerative medicine, tissue engineering, infectious diseases, COVID-19, influenza, HIV, tuberculosis, cholera

## Abstract

Mesenchymal stem cells (MSCs) are attractive in various fields of regenerative medicine due to their therapeutic potential and complex unique properties. Basic stem cell research and the global COVID-19 pandemic have given impetus to the development of cell therapy for infectious diseases. The aim of this review was to systematize scientific data on the applications of mesenchymal stem cells (MSCs) and MSC-derived extracellular vesicles (MSC-EVs) in the combined treatment of infectious diseases. Application of MSCs and MSC-EVs in the treatment of infectious diseases has immunomodulatory, anti-inflammatory, and antibacterial effects, and also promotes the restoration of the epithelium and stimulates tissue regeneration. The use of MSC-EVs is a promising cell-free treatment strategy that allows solving the problems associated with the safety of cell therapy and increasing its effectiveness. In this review, experimental data and clinical trials based on MSCs and MSC-EVs for the treatment of infectious diseases are presented. MSCs and MSC-EVs can be a promising tool for the treatment of various infectious diseases, particularly in combination with antiviral drugs. Employment of MSC-derived EVs represents a more promising strategy for cell-free treatment, demonstrating a high therapeutic potential in preclinical studies.

## 1. Introduction

Infectious diseases are a large group of diseases caused by the impact of various pathogenic or conditionally biological agents on the human body. Several types are distinguished depending on the origin of the pathogen: viral, bacterial, fungal, as well as infections caused by prions, protozoa, and parasites. There are many historic examples of the devastating consequences caused by infectious diseases (e.g., smallpox, plague, cholera, typhoid, influenza, etc.), which are called “plague diseases”. Despite the sanitary well-being and the achievements of modern medicine, it is naive to believe that humanity has defeated infectious diseases, and each of us is not at risk. Currently, epidemics of COVID-19, tuberculosis (TB), AIDS, malaria, measles, influenza, and other diseases are constantly active in the world. According to the World Health Organization (WHO), about 50% of the world’s population lives in conditions of constant threat of epidemics (www.who.int (accessed on 12 February 2019)).

There are several objective conditions for the development of infectious diseases: the active development of tourism [1,2,3], the increase in migration processes [4,5,6,7,8], returning and re-emerging diseases [9,10], as well as the likelihood of using pathogens of various infectious diseases as biological weapons [11,12,13]. Improved social and environmental conditions help with reducing the risk of contracting and spreading infectious diseases [14,15], but, paradoxically, as the living standards rise, the mortality from some of them also increases. For example, in the case of paralytic poliomyelitis or chicken pox, the severity of the infectious process complications (including pneumonia, acute neurological disorders, thrombocytopenia, chickenpox encephalitis with damage to the myelin sheaths of the brain and spinal cord, etc.) is directly correlated to the age of the patient [16,17]. The use of antibiotics and active immunization of the population have made it possible to defeat or take control of most of the infections, however, there are still many infectious diseases that cannot be treated (AIDS, a multidrug resistant form of TB, viral hepatitis C, prion infections, etc.), as well as leading to serious complications (COVID-19, influenza, etc.).

The active growth of basic stem cell research has given impetus to the development of translational medicine, which is grounded on the results obtained and promotes new treatments for various diseases. One of these areas is cell therapy, which is based on the use of cells and cellular secretome that can stimulate tissue regeneration, provide anti-inflammatory, immunomodulatory, and other therapeutic effects on the body [18,19]. Despite the fact that many biological mechanisms concerning the effect of MSCs on damaged tissues remain insufficiently studied, the possibility of using cell therapy for various diseases, including infectious diseases, both as a monotherapy or in combination with other agents, is currently being actively studied [20,21,22,23].

The global stem cell therapy market is projected to grow to USD 18.51 billion by 2026 at a compound annual growth rate of 9.8% (Appendix A). Such predictions are driven primarily by increased awareness of the therapeutic efficacy of stem cells, as well as the development of the infrastructure associated with obtaining and banking stem cells. The largest companies in this market are Anterogen Co., Ltd. (Seoul, Korea), Mesoblast Ltd. (Melbourne, Australia), Osiris Therapeutics Inc. (Columbia, MD, USA), AlloSource (Centennial, CO, USA), Cellular Engineering Technologies (Coralville, IA, USA), BIOTIME Inc. (Carlsbad, CA, USA), Astellas Pharma US Inc. (Northbrook, IL, USA), Vericel (Cambridge, MA, USA), RTI Surgical Inc. (Deerfield, IL, USA), and Takara Bio Company (Kusatsu, Tokyo).

At the moment, cell therapy is not yet widely used and distributed, which is associated with its high cost, as well as the personalized approach and the use of autologous cells. However, in recent years, there has been a significant demand for allogeneic cells due to a more affordable process for their cultivation and an increase in the commercialization of allogeneic therapy products. The most commonly used allogeneic stem cells in clinical research include: mesenchymal stem cells (MSCs) isolated from bone marrow (bone marrow-derived mesenchymal stem cells (BM-MSCs), adipose tissue (adipose tissue-derived mesenchymal stem cells (A-MSCs)), umbilical cord (umbilical cord blood-derived mesenchymal stem cells (UC-MSCs), and placenta (placenta-derived mesenchymal stem cells (P-MSCs). The data available support the safety of therapy with both autologous and allogeneic MSCs. Despite the fact that the evidence on the effectiveness of cell therapy is often preliminary, the great advantages of MSCs are still their weak immunogenic properties and the possibility of rapid application for the treatment of various diseases [24,25].

## 2. MSCs and MSC-Derived Extracellular Vesicles (MSC-EVs)

The concept of “adult” MSCs, first proposed by Kaplan, appeared in accordance with the concept of the cell origin in the embryonic mesoderm [26]. Despite that it does not strictly correspond to the biological definition of MSCs [27,28], this term is widely used by clinicians and scientists to this day [29]. Due to the use of various methods for obtaining and culturing stem cells, the discussion on the specific characteristics used to determine MSCs is rather conflicting. Since these cells can be isolated from almost any tissue, it has been suggested that MSCs from different sources may be sufficiently distinct to combine them into a single classification (Table 1).

BM-MSCs have a longer doubling time and age earlier compared to cells obtained from other sources [41,42]. Approximately 98–100% of cells remain viable when derived from adipose tissue (A-MSCs) compared to cell isolation from bone marrow [43]. A-MSCs secrete various cytokines and growth factors with anti-inflammatory, antiapoptotic, and immunomodulatory characteristics including vascular endothelial growth factor (VEGF), hepatocyte growth factor (HGF), and insulin-like growth factor (IGF), which are involved in angiogenesis and damaged tissues repair. Additionally, due to their immunomodulatory effects, A-MSCs are an excellent source for allogeneic transplants, since they do not express type II major histocompatibility complex (MHC class II) and the risk of transplant rejection is thus minimized [44]. A minimal risk of immune response was also observed with in vivo administration of allogeneic UC-MSCs. This property and ease of their preparation also make UC-MSCs suitable candidates for cell therapy [45]. P-MSCs express embryonic stem cell markers such as c-Kit, Oct-4, stage-specific embryonic antigen (SSEA-4), as well as markers that determine the sex of the donor Y-box 2. One of the main advantages when applying P-MSCs is their high proliferative properties and plasticity [46,47].

In 2006, the International Society for Cell Therapy (www.isctglobal.org) defined three minimum criteria for identifying MSCs: (1) adherence to plastic under standard culture conditions; (2) the ability to differentiate into osteogenic, adipocyte, and chondrogenic directions under appropriate cultivation conditions; and (3) phenotyping by the presence/absence of surface markers: ≥95% positive for CD105, CD73, and CD90; ≤5.2% negative for CD45, CD34, CD14/CD11b, CD79a/CD19, and HLA-DR [48]. These cell types (BM-MSCs, A-MSCs, UC-MSCs, and P-MSCs) share the minimal criteria defined by ISCT and have additional characteristics which are associated with their tissue specificity [49,50].

Due to their therapeutic potential and unique properties, MSCs are appealing for various fields of regenerative medicine [51,52], cancer therapy [53,54,55], and infectious diseases [20,56,57,58,59]. MSCs synthesize factors that can restore damaged tissues. It has recently been suggested that MSCs are able to modulate cellular autophagy in damaged tissues/organs. MSCs can affect the autophagy of immune cells involved in injury-induced inflammation, reducing their survival, proliferation, and level of inflammation. At the same time, MSCs promote survival, proliferation, and differentiation of endogenous adult or progenitor cells, thereby promoting tissue repair [60].

Initial preclinical data on the therapeutic efficacy of MSCs focused mainly on the regenerative and differentiating ability of cells. However, there is now increasing evidence that many, if not all, of the positive effects of MSCs are associated with the paracrine activity of cells and their secretome, which consists of EVs [61,62], soluble proteins, cytokines, chemokines, and growth factors [63,64,65], and not only with the integration of cells into the damaged area [66,67]. MSCs have been found to play an immunomodulatory role in numerous infection diseases through the production of soluble factors, and the transfer of EVs containing various molecules [68]. It has been established that MSC-EVs have the same immunomodulatory and anti-inflammatory and other effects as their parental cells and recapitulate a broad range of the therapeutic effects shown by MSC treatment. The functional differences between MSC and MSC-EVs are not significant [69]. However, there are different mechanisms underlying the interaction of various MSCs or MSC-EVs with immune cells. EVs derived from different types of MSCs have similar and unique characteristics (Table 2).

Currently, most clinical trials of MSC therapy for viral and bacterial infectious diseases have focused on patients who have not responded to traditional disease drug therapy as COVID-19, AIDS, and TB. However, the use of MSCs in therapeutic treatments still has many challenges. An increasing number of studies reveal that MSCs are highly heterogeneous with different multipotential properties, progenitors, and cell states. In addition, MSCs isolated from different sources exhibit distinct characteristics, known as tissue sources-associated heterogeneity [86,87,88]. Moreover, the intravenous administration of MSCs can lead to aggregation or clumping of cells in the vascular system and is associated with the risk of mutagenicity and oncogenicity [89,90].

The results of preclinical studies in vitro and in vivo show that MSC-EVs also exhibit significant therapeutic properties in many pathophysiological conditions of the body, restoring damaged organs and tissues [91,92,93] without the risks associated with direct cell engraftment (i.e., immunogenicity, tumorigenicity, and teratoma formation) [94,95]. The application of MSC-EVs in the treatment of diseases is a novel concept with particular advantages over the whole-cell therapy. MSC-EVs are well-tolerated and have low immu-nogenicity and also have a more stable membrane structure than MSCs. Another advantage of MSC-EVs over MSCs is the possibility of storing them for several weeks/months allowing their safe transportation and delayed therapeutic use [96]. These advantages of EVs provide broader prospects for disease treatment. However, the studying of the mechanism of EVs in the treatment of diseases is the primary connection to future clinical research.

### 2.1. Extracellular Vesicles (EVs)

EVs are heterogeneous vesicles surrounded by a lipid bilayer and secreted not only by MSCs, but also by all cell types. EVs mediate intercellular communication and are involved in many physiological and pathophysiological processes, including modulating immune responses, maintaining homeostasis, inflammation, angiogenesis, and others [97,98,99,100] (Figure 1).

Depending on the origin and size of EVs, they are divided into various subtypes: ectosomes, microvesicles, microparticles, exosomes, oncosomes, apoptotic bodies, etc. [101]. However, these EV subtypes are further characterized by different, often overlapping, definitions based primarily on vesicle biogenesis (cellular pathway, cellular or tissue identity, etc.) [102]. In order to avoid contradictions in definitions, the International Society for Extracellular Vesicles (ISEV) (www.isev.org (accessed on 3 May 2018)) proposed in 2018 to call a particle secreted by a cell an “extracellular vesicle” if its specific biogenetic origin cannot be demonstrated.

The regenerative potential of EVs is mainly explained by the regulation of apoptosis, cell proliferation, differentiation, angiogenesis, and inflammation [103]. The exact mechanisms underlying the therapeutic effects of EVs remain to be fully elucidated. However, several factors have proven to be promising contenders for the transfer of regenerative potential: microRNA (miRNA), messenger RNA (mRNA), and proteins. MSC-EVs of different origin are quite heterogeneous and have significant differences in the qualitative and quantitative composition of proteins, cytokines, nucleic acids, lipids, mRNA, microRNA, and other active components [104,105]. The paracrine action of EVs can be mediated through three mechanisms, including internalization, direct fusion, and ligand–receptor interaction with the target cell [106]. Using these pathways, EVs deliver various biomolecules and are involved in the inhibition and/or induction of signaling in target cells.

### 2.2. Application of Tissue Engineering Methods to Improve Therapeutic Effectiveness of MSCs and MSC-EVs

The rather low efficiency of cell therapy is one of the factors that significantly limits its use. The effectiveness of cell therapy is influenced by various factors: methods of administration, multiplicity, stability, efficiency of retention in the target tissue, heterogeneity, content of vesicles, etc. Positive effects of MSCs can be further enhanced in various ways, for example, by changing the method of cultivation (under conditions of hypoxia compared with normoxic conditions) [107,108], or the form of cultivation (3D or 2D culture) [109,110,111], as well as exposing cells to various influences (e.g., heat shock) [112], or genetic modifications [113]. It has been shown that pretreated MSCs demonstrate enhanced differentiation efficiency [114,115,116,117,118], improved paracrine functions [119], superior survival [120], and an enhanced ability of EVs to accumulate and remain in damaged tissue [121,122,123,124]. An additional approach that makes it possible to increase the efficiency of cell therapy is the use of various biomaterials [125,126], as well as the cultivation of cells in cell sheets [127,128,129].

Various methods can also be used to enhance the therapeutic efficacy of MSC-EVs: preconditioning of donor cells to increase the beneficial contents of EVs [130,131,132], using genetically modified cells [133] to change the composition of EVs [134,135], and dosing and multiple application [136]. However, each of these methods has its drawbacks. For example, the use of preconditioned media, unfortunately, does not give a high yield of MSC-EVs, which is a limiting factor for the use of cell-free therapy, while the genetic modification of cells and repeated administration of EVs can be potential risk factors for tumor growth.

The structural versatility of EVs provides an opportunity for surface modifications that can be performed using various methods, such as genetic engineering, chemical and physical methods, and microfluidic technologies [137]. The efficiency and scalability of methods for modifying EVs are critical in defining the scope for clinical application [138]. It is known that integrins are present on the surface membranes, and RGD peptides (Arg-Gly-Asp) have a high binding affinity for integrins [139,140]. RGD-modified EVs have been shown to exhibit increased targeting to blood vessels and represent a potential new therapeutic tool for angiogenesis therapy [141].

Another approach that contributes to the creation of a more stable therapeutic effect of EVs is the use of hydrogels. To date, there are several studies on the development of hydrogel scaffolds for loading EVs and evaluating the mechanisms of interaction between gels and EVs, which are still unclear [142,143]. It has been shown that a biocompatible self-assembling RGD hydrogel easily conjugates with EVs, and such constructs increase the therapeutic efficacy of MSC-derived vesicles for the treatment of kidneys [144], liver [145], and other organs [146]. Loading EVs with various drugs also enhances their therapeutic effect [147,148,149].

Research conducted to reveal the therapeutic potential of EVs, especially those that secrete MSCs, has proven to be significant for regenerative medicine. However, how EVs promote tissue regeneration and what drives their regenerative effect is still far from clear.

### 2.3. Mechanisms of Immunomodulatory Action of MSCs and MSC-EVs

MSCs are involved in innate and adaptive immunity and their immunomodulatory functions are manifested mainly when interacting with immune cells (T-cells, B-cells, natural killer (NK-cells), macrophages, monocytes, dendritic cells, and neutrophils) through the formation of intercellular contacts and implementation of paracrine activity [150]. By influencing the adaptive immune system, T-cells in particular, MSCs inhibit the differentiation of Th17, inducing the production of IL-10 and PGE2, as well as inhibiting IL-17, IL-22, and IFN-γ [151]. However, the mechanisms underlying the interactions between MSCs and Th17 lymphocytes have not yet been fully understood. In the innate immune system, MSCs interact with NK-cells, inhibit their proliferation with the help of IL-2, and induce cytotoxic activity, as well as the production of cytokines through the secretion of IDO and PGE2 [152].

The key role in the development of the immunomodulatory potential of MSCs is played by the interaction of cells with regulatory T-cells and monocytes. A-MSCs have been shown to regulate T-cell function by inducing suppressor T-cells and inhibiting the production of cytotoxic CD8+ T-cells, NK-cells, and proinflammatory cytokines including tumor necrosis factor-alpha (TFN-alpha), IFN-gamma, and IL-12. The secretion of A-MSCs of soluble factors such as IL-10, TGF-beta, and PGE2 renders cells immunosuppressive [44,153]. In this regard, A-MSCs have the strongest immunomodulatory effect compared to BM-MSCs and can become a better alternative for immunomodulatory therapy [154].

Through intercellular interactions, MSCs increase the survival of B-cells and promote their differentiation [155]. A-MSCs not only inhibit caspase-3-mediated B-cell apoptosis by up-regulating VEGF expression, but also inhibit proliferation by blocking the B-lymphocyte cell cycle in the G0/G1 phase by activating p38 protein kinase (MARK) [156]. In addition, MSCs prevent the death of neutrophils through an ICAM-1-dependent mechanism and exert a tissue protective effect [157]. Thus, the interaction of MSCs with immune cells contributes to a decrease in the inflammatory response, as well as the regeneration of damaged tissue.

## 3. Viral Infectious Diseases

Over the past decades, a huge number of experimental and clinical studies have been devoted to the use of cell therapy in the treatment of oncological, cardiovascular, neurodegenerative, and other diseases [158,159,160]. Basic stem cell research and the global COVID-19 pandemic have given rise to the development of cell therapy for infectious diseases, which currently stands at 121 registered clinical trials [161].

### 3.1. COVID-19

Coronavirus and other respiratory viruses are the leading cause of morbidity and mortality in acute lung injury (ALI) and acute respiratory distress syndrome. Although scientific advances have enabled rapid progress in understanding pathogenesis and developing therapeutic agents, stem cell therapy has recently found numerous applications in the treatment of viral infections.

Severe forms of the disease caused by COVID-19 are accompanied by increased activation of the immune system, which, in addition to antiviral protection, leads to a side effect—damage to lung tissue and other organs. To date, several studies have proposed the use of MSCs for the treatment of pneumonia caused by COVID-19 [162,163,164,165]. MSCs have been shown to reduce inflammation and suppress viral infection [166]. In the ALI mouse model, it was shown that, due to the anti-inflammatory effect, MSCs improve lung function, synthesizing the keratinocytes growth factor (KGF), VEGF, and HGF to restore damaged epithelial cells and lung tissues. IDO, TGF-ß, and granulocyte-macrophage colony-stimulating factor (G-CSF) act on macrophages, neutrophils, and T-cells (Figure 2). The main mechanism of action is probably to reduce the secretion of inflammatory factors [167].

Clinical intravenous administration of MSCs has shown an increase in the number of peripheral lymphocytes, hyperactivation of some types of T-cells as well as a decrease in the level of C-reactive protein [168]. A major factor in organ damage in severe COVID-19 cases is the cytokine storm. Due to their strong immunomodulatory ability, MSCs not only suppress the cytokine storm, but also promote the activation of the endogenous regenerative mechanism [169]. At the same time, MSC-EVs play an important role in the implementation of intercellular communication, since they are able to enter the bloodstream, pass through it for long distances and pass through histohematic barriers [170,171].

Several clinical studies have demonstrated the ability of MSC-EVs to reduce the level of inflammatory factors and increase immunity in various forms of COVID-19 (NCT04384445, USA; NCT04276987, China; NCT04491240, Russia). Two clinical trials are currently underway: one study group (NCT04276987) is investigating the efficacy of inhaled treatment of COVID-19 pneumonia using EVs derived from A-MSCs, and the second one (NCT04313647) is evaluating their safety and tolerability in healthy volunteers.

Due to their specific structure, various drugs can be introduced into MSC-EVs to use them as delivery systems [172] and one of the tools in the treatment of viral infection [173]. In addition, compared with other types of treatment, such as monoclonal antibody therapy, the economic costs of obtaining and using MSC-EV are significantly lower, which is important when using this method during a pandemic [28]. Ongoing clinical trials highlight the potential benefits of using both MSCs and MSC-EVs for the treatment of patients with COVID-19. However, further studies to evaluate and confirm their efficacy and safety are needed.

### 3.2. Flu

Due to the fact that infectious diseases of the respiratory organs caused by various viruses can occur like the common cold but also have severe acute respiratory syndromes [174], it is rather difficult to determine the specific agents involved in the infection [175]. Currently, influenza therapy mainly includes antibacterial and antiviral drugs.

Several studies on animal models infected with the influenza virus have shown a positive effect of the use of MSCs of various tissue origins [176,177,178,179,180]. Cocultivation of BM-MSCs with H5N1 virus-infected AECs inhibits their permeability under in vitro conditions. Possible mechanisms for this are related to the secretion of angiopoietin-1 (Ang1) and KGF by BM-MSCs [178]. In vivo experiments demonstrate that BM-MSCs have a significant anti-inflammatory effect by increasing the number of macrophages and releasing various cytokines and interleukins: IL-1 beta, IL-4, IL-6, IL-8, and IL-17 [180,181]. Similar anti-inflammatory effects have been shown using another model of lung injury caused by the H9N2 virus [179]. Intravenous injection of a suspension of BM-MSCs into virus-infected mice significantly attenuates virus-induced lung inflammation by reducing the levels of chemokines (GM-CSF, MCP-1, KC, MIP-1α, and MIG) and proinflammatory factors IL-1 alpha, IL-6, TNF-alpha, and IFN-gamma. Using an in vitro model of lung injury caused by the H5N1 virus, human UC-MSCs, through the secretion of Ang1 and HGF, had the same anti-inflammatory effect as BM-MSCs [182]. In one clinical study in patients with lung injury caused by the H7N9 influenza virus, the use of MSCs did not cause side effects and significantly increased their survival [183]. Despite the data indicating the therapeutic effect of MSCs in various preclinical models of lung injury, some studies have shown that the use of a suspension of MSCs with an antiviral drug was ineffective [184,185]. In addition, when using cell therapy, it is necessary to take into account the condition of the donor and recipient. It has been shown that when MSCs are administered to a patient with ongoing disease, cells can become infected with the influenza virus, and transplantation of BM-MSCs from influenza virus-infected donors, in turn, can also transmit the infection to recipients. Thus, when using cell therapy in the treatment of pulmonary influenza, it is imperative to take into account these factors and observe safety.

### 3.3. AIDS

The human immunodeficiency virus (HIV) is caused by a retrovirus of the lentivirus genus. It affects cells of the immune system that have CD4 receptors on their surface: T-helpers, monocytes, macrophages, Langerhans cells, dendritic cells, and microglial cells. As a result, the work of the immune system is inhibited and the syndrome of acquired immune deficiency (AIDS) develops, the patient’s body loses the ability to defend itself against infections and tumors. Despite the fact that the first case of AIDS was discovered almost 27 years ago, it is still not possible to effectively control the AIDS pandemic [186]. Of the 35 million people living with HIV infection, a fraction survives thanks to antiretroviral therapy, but in the absence of it, death occurs on average 9–11 years after infection. There are currently three known cases of a cure for the virus. In the medical literature, they appear under the names “Berlin”, “London”, and “Sao Paulo” patients [187,188].

Recently, a new strategy for the treatment of HIV and AIDS using stem cells, in particular BM-MSCs, has emerged [189]. According to the data, circulating replicative HIV remains the most serious threat to effective AIDS therapy. The main therapy strategy is aimed at reducing the number of replicating virus particles. As a result of its application, the destruction of HIV circulating in the blood occurs with the help of erythrocytes integrated with the CD4 receptor and chemokine receptors, which selectively bind circulating HIV particles [190,191,192,193].

One of the most interesting studies focused on the use of MSCs to increase antiviral immune activity and minimize the amount of virus. It has been shown that the administration of MSCs, even in the absence of antiviral drugs, can enhance the host’s antiviral response due to the restoration of lymphoid follicles and mucosal immunity, all of which become the target of the virus at an early stage [194]. The results of scientific and clinical studies provide an appropriate scientific basis for the future use of MSCs in the treatment of HIV and other infectious diseases. Researchers are still developing comprehensive and effective treatments for AIDS and related conditions.

Cell-based therapies initially were reserved to the most severely affected patients with viral infectious diseases (COVID-19, flu, and AIDS) and most clinical trials were also focused on them (Table 3).

The application of cell and vesicles therapy in most of the clinical trials resulted in symptomatic relief and treatment success. However, in order to ensure the widespread clinical implementation of MSC-based therapy, there are many challenges that need to be resolved (stages of the disease, clinical indicators, gender and age of patients, the source and age of MSCs, etc.).

## 4. Bacterial Infectious Diseases

### 4.1. Tuberculosis

Tuberculosis (TB) is one of the 10 leading causes of death worldwide. According to WHO data for 2021, over 9.9 million people worldwide became infected and about 1.3 million people died from TB [218]. The emergence of the COVID-19 pandemic has severely disrupted global TB prevention and control [219,220]. Nearly half a million people suffer from the rifampin-resistant TB strain, of which 78% are multidrug-resistant. In this regard, the actual direction is the search for fundamentally new approaches in the treatment of resistant TB, among which a certain place is occupied by MSC therapy.

Once in the lower respiratory tract, mycobacteria (Mycobacterium tuberculosis (µTb)) are mainly absorbed by macrophages. In this case, the resulting inflammatory reaction causes a large number of immune cells (monocytes, dendritic cells, neutrophils, and T-lymphocytes) to be attracted to the infected area, resulting in the formation of tuberculous granuloma (TG), which is a pathological sign of TB [221,222]. TG formation is a key event in preventing the spread of infection, and the period during which µTb are able to avoid the host’s immune response and remain dormant can be decades [223]. Numerous studies show that MSCs are involved in the formation and development of TG. It was found using CD29 as a marker that the cells are in a cluster with acid-resistant bacteria and are distributed in the TG area. In the pathogenesis of TB, MSCs, on the one hand, are able to inhibit the T-cell response through the synthesis of nitric oxide (NO) and, thereby, reduce the immune response, and on the other hand, NO itself can inhibit the growth of µTb and limit their proliferation within TG. Thus, it can be assumed that the formation of TG is associated precisely with this mechanism [224]. It has been shown that MSCs are able to regulate and limit the growth of µTb [225] using scavenger receptors for this [226,227]. Whether MSCs can influence the growth of µTb in any other way is still a question that still needs further study.

It has been shown that MSCs are natural host cells of latent µTb infection. In addition, recent research found that MSCs exist in the lungs and extrapulmonary tuberculosis granuloma. After infection with MSCs, the metabolic activity of µTb in cells becomes low, and, thus, they gradually acquire resistance to antituberculosis drugs [228,229]. Thus, along with immune cells, MSCs can not only provide a niche for dormant µTb but also limit their growth to a certain extent and participate in the emergence and development of TB.

The incidence of TB largely depends not on primary or secondary infection but on the reactivation of the dormant form of TB against the background of the emerging immunodeficiency [230]. In this regard, in recent years, therapy methods aimed at increasing infection control, reducing inflammation by modulating the immune response, and reducing tissue damage have become widespread [231]. Immunomodulatory properties and the ability to replace or repair damaged tissues make MSCs ideal candidates for the treatment of both pulmonary and extrapulmonary TB [232]. A number of studies have shown that the therapeutic potential of MSCs is associated with the antibacterial activity of cells directed against various pathogens (Escherichia coli, Pseudomonas aeruginosa, Staphylococcus aureus, Streptococcus pneumonia) through the secretion of antimicrobial peptides [233,234,235]. However, it is not yet known whether MSCs affect µTb growth in the same way.

MSCs and MSC-EVs have a wide range of immunomodulatory effects on various cells of the immune system: they promote the function of regulatory T cells (Treg and Th2) [236,237], inhibit the release of IFN-gamma, regulate the balance of Th1/Th2 [238], promote polarization of macrophages from M1 to M2 by expression of IDO and activation of CD39 and CD73/adenosine signaling pathways [237,238,239], and inhibit activation and promote B cell transformation [240,241,242]. In addition, MSCs are able to regulate the survival of the alveolar epithelium by secreting factors KGF and HGF that protect cells from apoptosis [243].

Previously, we showed that intravenous administration of MSCs results in accumulation and retention of MSCs in µTb-affected rabbit kidney tissue, due to which the cells are able to reduce the level of the inflammatory response and enhance the process of tissue repair [244]. A decrease in the level of expression and synthesis of hydroxyproline, collagen Types I and III leads to a decrease in fibrosis, restoration of damage, and prevention of pulmonary edema [245,246,247].

A number of studies have shown that, at low concentrations, MSCs can inhibit the activation of lymphocytes [248]. Thus, the ratio of the number of MSCs and immune cells can be a turning point for inhibiting or activating the immune response. Overall, the results obtained with MSCs in vivo are encouraging, but the safety and efficacy of MSCs in the treatment of TB remains to be confirmed.

### 4.2. Cholera

Vibrio cholerae (VCh) is the causative agent of cholera, which is commonly associated with a high infection rate, mortality, and a major public health problem in many parts of the world [249,250]. According to WHO data, each year there are from 1.3 to 4 million cases of cholera, and 100,000 to 130,000 deaths worldwide due to cholera per year. The emergence of multidrug-resistant VCh strains in developing countries is of great concern [251,252]. The high mortality rate and the lack of effective antimicrobials necessitate the development of new effective approaches for the treatment of drug-resistant strains. Various vaccines have been developed (Dukoral, Shanhol, and Euvichol), but none provide complete long-term protection and are not approved for use in children under 1 year of age. Inflammation caused by the interaction of Vibrio cholerae with epithelial cells is considered as the main cause of the spread of bacteria in the gastrointestinal tract and the progression of its consequences. One of the effective therapeutic approaches to treatment is to reduce the level of inflammatory cytokines caused by VCh infection.

MSCs exert their antibacterial properties through the synthesis of compounds such as antimicrobial peptides (hCAP18/LL-37), which control the growth and reproduction of bacteria. One study using a neonatal mouse model showed the immunomodulatory effect of a medium conditioned with MSCs supplemented with LPS (lipopolysaccharide necessary to protect the body from VCh) [253], a decrease in the level of the inflammatory response and the induction of the production of vibriocidal antibodies that protect against VCh. In addition, MSCs have been shown to be effective in the treatment of bacterial sepsis [254,255].

A-MSCs show dual effects on inflammatory response and epithelial barrier integrity by reduction of bacterial attachment and increasing bacterial internalization. On the one hand, A-MSCs reduce bacterial adhesion and colony formation by secreting various antimicrobial peptides (including IDO, and TIMP). A decrease in the rate of bacterial adhesion, in turn, leads to a decrease in the expression of chloratoxin and an increase in the secretion of IL-6, which has a positive effect on maintaining the integrity of the epithelial barrier. On the other hand, increased bacterial internalization by cells stimulates the inflammatory reactions. An increase in the level of expression of the proinflammatory genes TNF-alpha, IL-1beta, and IL-8 leads to an increase in the level of cytokines, induction of apoptosis, and degradation of the tight junction between epithelial cells. Thus, A-MSCs are able to exert different effects on the inflammatory response and the integrity of the epithelial barrier by reducing bacterial adhesion and enhancing bacterial internalization. The probable reason for this effect is the high level of MSC expression of matrix metalloproteinases and tissue inhibitor of proteinases (TIMP), as well as other antibacterial peptides [256]. Therefore, it is recommended that future studies focus on the protective effects of MSCs’ secretome.

It can be assumed that the reduction of bacterial internalization may also become an appropriate therapeutic approach to limit the inflammatory reactions caused by VCh, while it is more efficient to use MSC-EVs as a therapeutic agent instead of intact cells.

Currently, the evaluation of the effectiveness of cell therapy for bacterial infectious diseases is carried out mainly in vitro and in vivo conditions. There are few clinical studies on this topic (Table 4).

## 5. Conclusions

The therapeutic use of MSCs is not an unrealistic goal, as the cells offer a promising treatment option for a number of diseases. Using MSC-EVs instead of cells seems to be a promising strategy for cell-free treatment, as it allows to solve various problems associated with cell administration. Nevertheless, for clinical use, a preliminary assessment of the safety, efficacy, and long-term results of using both various types of MSCs, regardless of the source of their production, and MSC-EVs, first in animal models and then in preclinical trials, is necessary. At present, studies are being actively carried out on the selection and establishment of optimal therapeutic doses and the frequency of administration of cells and vesicles, optimal methods for their management, assessment of cellular and vesicular heterogeneity, etc. In addition, it is known that the properties of MSCs change significantly under inflammatory or anti-inflammatory stimuli, and therefore, it remains to be seen how variability affects the immunomodulatory effects induced by cells and to establish which subpopulations of cells or extracellular vesicles are the most therapeutically effective. Although clinical research on MSCs is still in its infancy, there is great hope that MSCs and MSC-EVs will become promising tools for future clinical applications in the treatment of infectious diseases.

## Figures and Tables

**Figure 1 bioengineering-09-00662-f001:**
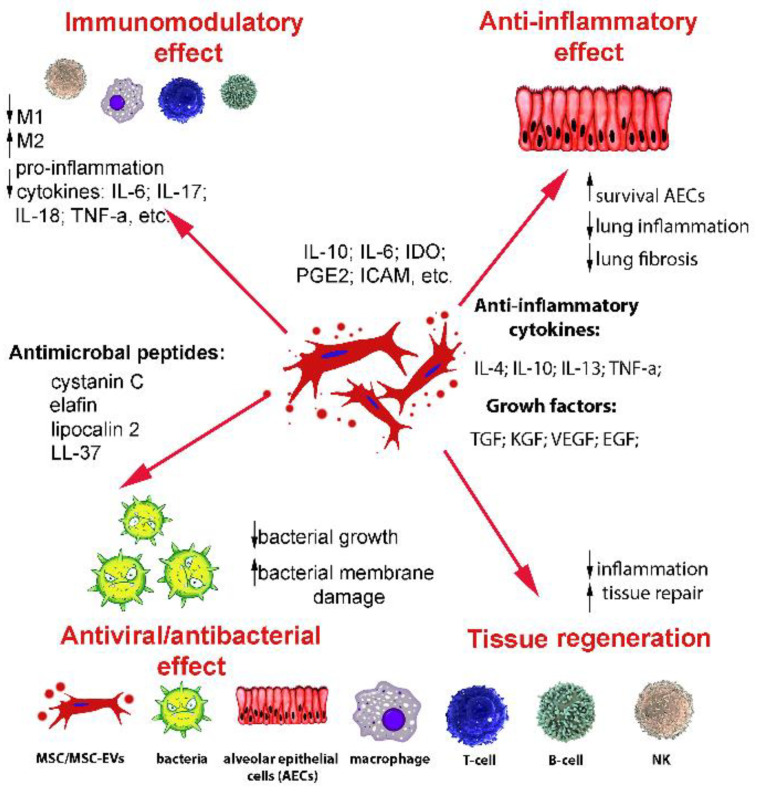
Molecules released by MSC-EVs. The MSC-EVs contain cytokines, growth factors, and other active molecules that maintain the modulation of the immune response, have anti-inflammatory, antiviral/antibacterial effects, and promote epithelial repair and tissue regeneration. MSCs, mesenchymal stem/stromal cells; EVs, extracellular vesicles; AECs; alveolar epithelial cells; macrophage; T-cell; B-cell; NK, natural killer cell.

**Figure 2 bioengineering-09-00662-f002:**
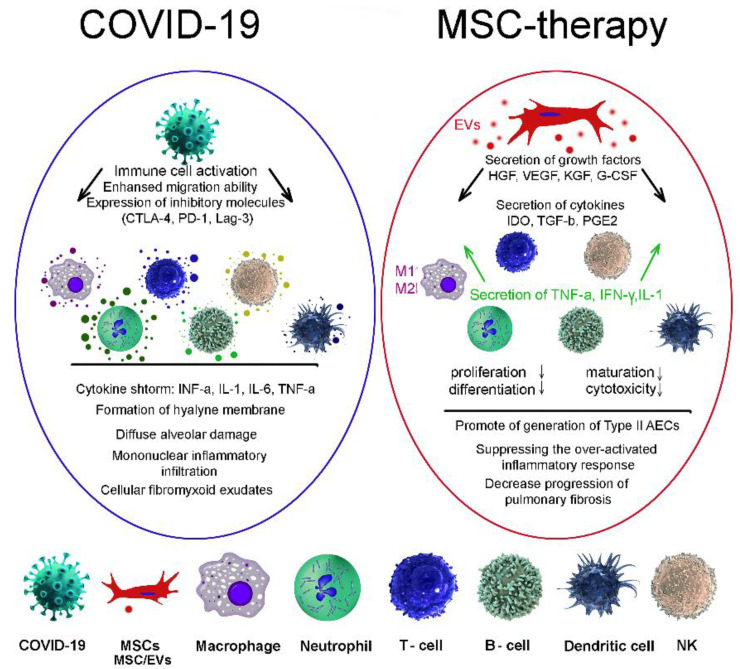
MSC therapies for treatment of coronavirus-induced lung injury. COVID-19, coronavirus; MSCs, mesenchymal stem/stromal cells; EVs, extracellular vesicles; macrophage; neutrophil; T-cell; B-cell; dendritic cell; NK, natural killer cell.

**Table 1 bioengineering-09-00662-t001:** Commonly used sources of MSCs.

	Source	Abbreviation	Proliferation Rate	Doubling Time	Immunogenicity	MSCs Phenotype	References
1	Bone Marrow	BM-MSCs	Lowest	40 h	Medium	Stro-1, CD271, SSEA-4, CD146	[30,31,32,33,34,35,36]
2	Adipose Tissue	A-MSCs	Higher	5 days	High	CD271, CD146	[33,34,35,36]
3	Umbilical Cord	UC-MSCs	Medium	30 h	High	CD146	[35,36,37,38]
4	Placenta	P-MSCs	High	36 h	High	c-Kit, Oct-4, SSEA-4, Y-box 2	[39,40]

**Table 2 bioengineering-09-00662-t002:** Mechanisms of MSCs and MSC-EVs influence on immune cells.

	Abbreviation	Immune Cell	Mechanism	Effect	Reference
1	BM-MSC-EVs	CD4 + T cell	EVs-encapsulating miR-23a-3p and post-transcriptionally regulated TGF-beta receptor 2 in T cells	suppressive Th1 differentiation	[70]
2	BM-MSC-EVs	T cell	increasing IL-10 and TGF-beta	promote T cells apoptosis and inhibit proliferation	[71]
3	UC-MSCs	T cell	through the COX2/PGE2/NF-kB signaling pathway	inhibiting T cell proliferation and DC differentiation	[72]
4	AD-MSC	T cell	through regulating TGF-beta and PGE2	regulate the Th17/Treg balance	[73]
5	BM-MSC	B cell	inhibition of BAFF production	suppress the excessive activation of B-cells	[74]
6	BM-MSC-EVs	B cell	targeting PI3K/AKT signaling pathway	inhibit activation of B cell	[75]
7	BM-MSC	B cell	increased expression of CCL2 by CCL2-MST1-mTOR-STAT1 mediated metabolic signaling pathway	prevent inhibition differentiation, proliferation, and antibody secretion of B-cell	[76]
8	BM-MSC-EVs	DCs	expression of anti-inflammatory factors (TGF-beta 1 and IL-10) and reduce the generation of proinflammatory cytokines (L-6 and IL-12p70)	attenuate DCs maturation and function	[77]
9	G-MSC	NK cell	regulating IDO and PGE2	inhibit the activity of NK cells	[78]
10	BM-MSCs	NK cell	inhibit IL-12 and IL-21	suppression NK cell proliferation but increase IFN-gamma and IFN-alpha production	[79]
11	UC-MSC	macrophage	regulating macrophage metabolic pathways	affect M1/M2 balance	[80]
12	BM-MSC-EVs	macrophage	down-regulating IL-23 and IL-22	enhances the anti-inflammatory phenotype of macrophages, promoting inflammation remission	[81]
13	AD-MSCs	macrophage	down-regulating IL-23 and IL-22	toward M2 phenotype polarization	[82]
14	BM-MSC-EVs	macrophage	through miR-223/pKNOX1 pathway	promoting macrophages differentiation toward M2	[83]
15	BM-MSC-EVs	macrophage	through TLR4/NF-kB/PI3K/Akt signaling cascade	toward M2 phenotype polarization	[84]
16	UC-MSC-EVs	macrophage	increased the proportion of M2 macrophage polarization	attenuate DAH induced inflammatory responses and alveolar hemorrhage	[85]

MSC, mesenchymal stem cells; EVs, extracellular vesicles; BM-MSC, bone marrow-derived mesenchymal stem cells; BM-MSC-EVs, bone marrow MSC-derived extracellular vesicles; UC-MSC, umbilical cord-derived mesenchymal stem cells; AD-MSC, adipose-derived MSC; G-MSC, gingiva-derived mesenchymal stem cells; DCs, dendritic cells; IL, interleukin; IDO, indoleamine 2,3-dioxygenase; PGE2, prostaglandin E2; Th1, T-helper 1; Treg, regulatory T; IFN, interferon; TGF-beta 1, transforming growth factor beta 1; DAH, diffuse alveolar hemorrhage; Tfh, T follicular helper; IL-10, interleukin 10; TLR4, toll-like receptor 4; Th17, T-helper 17.

**Table 3 bioengineering-09-00662-t003:** MSCs and MSC-EVs based clinical trials of the viral infection diseases.

	Study Title	Abbreviation	Viral Infectious Diseases	Status	Country	Description	Reference
1	Bone Marrow-Derived Mesenchymal Stem Cell Treatment for Severe Patients With Coronavirus Disease 2019 (COVID-19)	BM-MSCs	COVID-19	Phase 2	China	Conventional treatment plus BM-MSCs (1 × 10^6^ cells/kg body weight intravenously	[195]
2	Treatment of Severe COVID-19 Pneumonia with Allogeneic Mesenchymal Stromal Cells (COVID_MSV)	BM-MSCs	COVID-19	Phase 2	Spain	IV injection of 1 × 10^6^ cells/kg diluted in 100 mL saline	[196]
3	Mesenchymal Stem Cell Therapy for SARS-CoV-2-related Acute Respiratory Distress Syndrome	BM-MSCs/ BM-MSC-EVs	COVID-19	Phase 3	Iran	Two doses of MSCs 1 × 10^8^ at Day 0 and Day 2 plus conventional treatment	[197]
4	Cellular Immuno-Therapy for COVID-19 Acute Respiratory Distress Syndrome—Vanguard (CIRCA-19)	BM-MSCs	COVID-19	Phase 1	Canada	IV administration	[198]
5	Mesenchymal Stromal Cells for the Treatment of SARS-CoV-2 Induced Acute Respiratory Failure (COVID-19 Disease)	BM-MSCs	COVID-19	Early Phase 1	USA	1 × 10^8^ cells/kg body weight intravenously	[199]
6	A Pilot Clinical Study on Inhalation of Mesenchymal Stem Cells Exosomes Treating Severe Novel Coronavirus Pneumonia	BM-MSC-EVs	COVID-19	Phase 1	China	5 times aerosol inhalation of MSC-EVs (2 × 10^8^/3 mL at Day 1, Day 2, Day 3, Day 4, Day 5)	[200]
7	Evaluation of Safety and Efficiency of Method of Exosome Inhalation in SARS-CoV-2 Associated Pneumonia. (COVID-19EXO)	BM-MSC-EVs	COVID-19	Phase 1 Phase 2	Russia	Twice a day during 10 days inhalation of 3 mL special solution contained 0.5–2 × 10^10^ of EVs.	[201]
8	Mesenchymal Stem Cells (MSCs) in Inflammation-Resolution Programs of Coronavirus Disease 2019 (COVID-19) Induced Acute Respiratory Distress Syndrome (ARDS)	BM-MSCs	COVID-19	Phase 2	Germany	Infusion of allogeneic bone marrow-derived human mesenchymal stem (stromal) cells	[202]
9	Safety and Efficacy of Mesenchymal Stem Cells in the Management of Severe COVID-19 Pneumonia (CELMA)	UC-MSCs	COVID-19	Phase 2	USA	1 × 10^6^ cells/kg body weight intravenously	[203]
10	Therapy for Pneumonia Patients Infected by 2019 Novel Coronavirus	UC-MSCs	COVID-19	With- drawn	China	0.5 × 10^6^/kg body weight suspended in 100 mL saline intravenously at Day 1, Day 3, Day 5, Day 7	[204]
11	Use of UC-MSCs for COVID-19 Patients	UC-MSCs	COVID-19	Phase 2	USA	Conventional treatment plus UC-MSCs (1 × 10^8^/kg body weight intravenously	[205]
12	Study of Human Umbilical Cord Mesenchymal Stem Cells in the Treatment of Severe COVID-19	UC-MSCs	COVID-19	Not yet recruiting	China	4 times of UC-MSCs (0.5 × 10^6^ UC-MSCs cell/kg body weight intravenously at Day 1, Day 3, Day 5, Day 7)	[206]
13	Clinical Research of Human Mesenchymal Stem Cells in the Treatment of COVID-19 Pneumonia	UC-MSCs	COVID-19	Phase 2	China	1 × 10^6^ UC-MSCs/kg suspended in 100 mL saline	[207]
14	Autologous Adipose-derived Stem Cells (AdMSCs) for COVID-19	A-MSCs	COVID-19	Phase 2	USA	3 doses of 2 × 10^6^ cells through IV every 3 days	[208]
15	Battle Against COVID-19 Using Mesenchymal Stromal Cells	A-MSCs	COVID-19	Phase 2	Spain	Two serial doses of 1.5 × 10^6^ cells/kg	[209]
16	Clinical Trial to Assess the Safety and Efficacy of Intravenous Administration of Allogeneic Adult Mesenchymal Stem Cells of Expanded Adipose Tissue in Patients With Severe Pneumonia Due to COVID-19	A-MSCs	COVID-19	Phase 2	Spain	Two doses of 8 × 10^6^ A-MSCs	[210]
17	ASC Therapy for Patients with Severe Respiratory COVID-19	A-MSCs	COVID-19	Phase 2	Denmark	1 × 10^8^ cells/kg diluted in 100 mL saline	[211]
18	Zofin (Organicell Flow) for Patients With COVID-19	MSC-EVs	COVID-19	Phase 1	USA	Zofin with 1ml, containing 2–5 × 10^11^ EVs/mL in addition to the Standard care.	[212]
19	Umbilical Cord Mesenchymal Stem Cells for Immune Reconstitution in HIV-infected Patients	UC-MSCs	HIV/AIDS	Phase 2	China	High and low doses of MSCs (at 0, 4, 12, 24, 36 and 48 week since the onset of treatment)	[213]
20	Treatment with MSC in HIV-infected Patients with Controlled Viremia and Immunological Discordant Response	A-MSCs	HIV/AIDS	Phase 1 Phase 2	Spain	Intravenous infusion of 4 doses of A-MSCs (1 × 10^6^ cells/kg, weeks 0-4-8-20).	[214]
21	A Tolerance Clinical Study on Aerosol Inhalation of Mesenchymal Stem Cells Exosomes In Healthy Volunteers	MSC-EVs	Healthy Volunteers	Phase 1	China	Aerosol inhalation of MSC-EVs	[215]
22	Using Human Menstrual Blood Cells to Treat Acute Lung Injury Caused by H7N9 Bird Flu Virus Infection	MSCs	H7N9 Bird Flu Virus Infection	Phase 1 Phase 2	China	1~10 × 10^7^ cells/kg infusion frequency: 2 times a week, 2 weeks for infusion	[216]
23	Regenerative Medicine for COVID-19 and Flu-Elicited ARDS Using Lomecel-B (RECOVER)	MSCs	ARDS COVID-19	Phase 1	USA	1 × 10^8^ cells/kg on Day 0	[217]

MSC, mesenchymal stem cells; EVs, extracellular vesicles; MSC-EVs, MSC-derived extracellular vesicles; BM-MSC, bone marrow-derived mesenchymal stem cells; AD-MSC, adipose-derived MSC; UC-MSC, umbilical cord-derived mesenchymal stem cells.

**Table 4 bioengineering-09-00662-t004:** MSC- and MSC-EV-based clinical trials of the bacterial infection diseases.

	Study Title	Abbreviation	Bacterial Diseases	Status	Country	Description	Reference
1	Effectivity of Local Implantation of the Mesenchymal Stem Cell on Vertebral Bone Defect Due to Mycobaterium Tuberculosis Infection (Clinical Trial)	MSCs	Extrapulmonary tuberculosis	Phase 2	Indonesia	3 × 10^7^ cells/kg diluted in 2 mL 0.9% NaCl intravenously	[257]
2	Systemic Transplantation of Autologous Mesenchymal Stem Cells of the Bone Marrow in the Treatment of Patients With Multidrug-Resistant Pulmonary Tuberculosis	MSCs	Tuberculosis; multidrug resistant, extensive drug resistant	Completed	Russia	Not stated	[258]
3	Autologous Mesenchymal Stromal Cell Infusion as Adjunct Treatment in Patients With Multidrug and Extensively Drug-Resistant Tuberculosis: An Open-Label Phase 1 Safety Trial.	BM-MSCs	Tuberculosis; multidrug resistant, extensive drug resistant	Phase 1	Belarus	1 × 10^7^ cells/kg diluted in saline	[259]
4	Effectiveness of a Novel Cellular Therapy to Treat Multidrug-Resistant Tuberculosis.	BM-MSCs	Tuberculosis; multidrug resistant, extensive drug resistant	Phase 1	Belarus	1 × 10^7^ cells/kg diluted in saline	[260]

MSC, mesenchymal stem cells; BM-MSC, bone marrow-derived mesenchymal stem cells.

## Data Availability

Not applicable.

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
