# Peer review of "Mesenchymal Stem Cells and MSCs-Derived Extracellular Vesicles in Infectious Diseases: From Basic Research to Clinical Practice"

_bioengineering, 2022, doi:10.3390/bioengineering9110662_

Round 1

Reviewer 1 Report

Reviewer’s Comments

MSC exosomes are growing field of research due to their immunomodulatory and regenerative functions. This manuscript well highlighted an important function and potential applications of MSC exosomes and their anti-inflammatory and wound healing effects in various infectious diseases. We found some sentence construction in the latter half could be improved for better understanding of the readers. The details are below.

·      Please include your reference(s) for lines 45-48, 48-52, 184-187, 312-315.

·      It would be better to also place the reference(s) in Table 1

·      In line 50, do you mean life span instead of living standards since you cited the proportionality of chickenpox severity with age? 

·      In line 70, please check your figure reference as it seems to be not working. 

·      Please check and correct typographical errors

o   In line 97: “the on” after the comma.

o   In figure 2: “glicolytic”.

o   In line 255:  excess “(“.  

o   Figure 3: “mononuclea”

·      Lines 118-123 (criteria for MSC)

o   Why did you include these criteria? Did the 4 types of cells from table 1 all satisfy the criteria provided before they were classified as MSCs? 

o   It is now said that these criteria are not appropriate to define a bonafide MSC population due to changes in cell characteristics especially with the in vitro assays being used. What are your thoughts about that?

·      Please clarify the sentence in line 255. Did you mean that the MSCs synthesized factors that can restore damaged tissues?

·      In line 266 and 338, what type of MSC was administered? BM-MSC/A-MSC? 

·      In line 332, result of application of what? Application of BM-MSCs?

·      In line 367, what does SR mean?

·      In line 370, what do you mean by infection with MSC?

·      In line 408, is the death count 100-130 or 130,000 per year?

·      In line 435, please elaborate on the implications of A-MSC enhancing bacterial internalization despite reducing bacterial adhesion. 

·      A summary table of the different applications and known mechanism of MSC and MSC-EV in the different infectious conditions is recommended.

·      Please remove sentence from template in line 459. 

Author Response

Reviewing: 1

MSC exosomes are growing field of research due to their immunomodulatory and regenerative functions. This manuscript well highlighted an important function and potential applications of MSC exosomes and their anti-inflammatory and wound healing effects in various infectious diseases. We found some sentence construction in the latter half could be improved for better understanding of the readers. The details are below.

COMMENT 1: Please include your reference(s) for lines 45-48, 48-52, 184-187, 312-315.

ANSWER 1: References were included for these lines.

COMMENT 2: It would be better to also place the reference(s) in Table 1.

ANSWER 2: References were inserted into the Table 1.

COMMENT 3: In line 50, do you mean life span instead of living standards since you cited the proportionality of chickenpox severity with age? 

ANSWER 3: For example, in the case of paralytic poliomyelitis or chicken pox, the severity of the infectious process complications (i.e., chicken pneumonia, acute neurological disorders, thrombocytopenia, chickenpox encephalitis with damage to the myelin sheaths of the brain and spinal cord and etc.) is directly correlate to the age of the patient.

COMMENT 4: In line 70, please check your figure reference as it seems to be not working. 

ANSWER 4: Reference was corrected and Figure 1 was replaced as Supplement.

COMMENT 5: Please check and correct typographical errors.

ANSWER 5: We have corrected the typographical errors throughout the manuscript.

COMMENT 6: In line 97: “the on” after the comma. Despite that it does not strictly correspond to the biological definition of MSCs [10,11], the on term is widely used by clinicians and scientists to this day [12].

ANSWER 6: We have clarified this in manuscript.

COMMENT 7: In figure 2: “glicolytic”.

ANSWER 7: Figure 2 was corrected.

COMMENT 8: In line 255:  excess “(“.  

ANSWER 8: We have changed the word to ‘numerous’. Although scientific advances have enabled rapid progress in understanding pathogenesis and developing therapeutic agents, stem cell therapy has recently found numerous applications in the treatment of viral infections.

COMMENT 9: Figure 3: “mononuclea”

ANSWER 9: ‘mononuclea’ was corrected as mononuclear.

COMMENT 10: Lines 118-123 (criteria for MSC). Why did you include these criteria? Did the 4 types of cells from table 1 all satisfy the criteria provided before they were classified as MSCs? 

ANSWER 10: There are no single specific markers that can be used to identify multipotent MSCs. MSC are characterized by a set of minimal criteria defined by the International Society for Cell & Gene Therapy (ISCT) [48, Dominici, et al, 2006]. These cell types (BM-MSCs, A-MSCs, UC-MSCs and P-MSCs) share minimal criteria defined by ISCT and have additional characteristics, which are associated with their tissue specificity [49, Mushahary, et al, 2018; 50, Nagamura-Inoue, et al., 2014].

COMMENT 11: It is now said that these criteria are not appropriate to define a bonafide MSC population due to changes in cell characteristics especially with the in vitro assays being used. What are your thoughts about that?

ANSWER 11: There are many factors that influence the properties, characteristics and therapeutic potential of MSCs. Therefore, a standard protocol for the in vitro culture of MSCs needs to be formulated. Furthermore, for clinical applications, a completely different and more advanced approach to characterize the cells and cell products is needed.

COMMENT 12: Please clarify the sentence in line 255. Did you mean that the MSCs synthesized factors that can restore damaged tissues?

ANSWER 12: It has recently been suggested that MSCs are able to modulate cellular autophagy in damaged tissues/organs. MSCs can affect the autophagy of immune cells involved in injury-induced inflammation, reducing their survival, proliferation, and level of inflammation. At the same time, MSCs promote survival, proliferation, and differentiation of endogenous adult or progenitor cells, thereby promoting tissue repair [60, Ceccariglia, et al., 2020].

COMMENT 13: In line 266 and 338, what type of MSC was administered? BM-MSC/A-MSC? 

ANSWER 13: Unfortunately, here we can not indicate specifically the type of MSCs. “A pilot trial of intravenous MSC transplantation was performed on seven patients with COVID-19 infected pneumonia. The study was conducted in Beijing You An Hospital, Capital Medical University, China, and approved by the ethics committee of the hospital (LL-2020-013-K)”.

COMMENT 14: In line 367, what does SR mean?

ANSWER 14: ‘SR’ stands for scavenger receptors. Scavenger receptors (SR) constitute a large family of cell-surface receptors that are diverse in their structure and biological functions (e.g., scavenging, phagocytosis, endocytosis, adhesion, and signaling) [227, Alquraini et al, 2020].

COMMENT 15: In line 370, what do you mean by infection with MSC?

ANSWER: It has been shown that MSCs are natural host cells of latent Mtb infection. In addition, recent research found that MSCs exist in the lungs and extrapulmonary tuberculosis granuloma [228, Jain, et al., 2020].

COMMENT 16: In line 408, is the death count 100-130 or 130,000 per year?

ANSWER 16: The sentence was corrected. According to WHO data each year there are from 1.3 to 4.0 million cases of cholera, and 100.000 to 143.000 deaths worldwide due to cholera per year.

COMMENT 17: In line 435, please elaborate on the implications of A-MSC enhancing bacterial internalization despite reducing bacterial adhesion. 

ANSWER 17: The intestinal epithelial cells (Caco-2) were co-cultured with A-MSC and the immunomodulatory and antibacterial effects of A-MSCs were investigated in vitro. In this study, A-MSCs show dual effects on inflammatory response and epithelial barrier integrity by reduction of bacterial attachment and increasing bacterial internalization. On the one hand, A-MSCs reduced bacterial attachment and colonization by secretion of various antimicrobial peptides (including IDO and TIMP). Decreased bacterial adhesion led to a decrease in the expression of chloratoxin, increased secretion of IL-6, which should have positive effect on the integrity of the epithelial barrier. On the other hand, increased bacterial internalization by A-MSCs stimulates the inflammatory responses. Therefore, it is recommended that future studies focus on the protective effects of MSCs secretome.  

COMMENT 18: A summary table of the different applications and known mechanism of MSC and MSC-EV in the different infectious conditions is recommended.

ANSWER 18: New Table 2 entitled “Mechanisms of MSCs and MSC-EVs influence on immune cells” was added.

COMMENT 19: Please remove sentence from template in line 459. 

ANSWER 19: The sentence was removed.

Reviewer 2 Report

Yudintceva et al review the scientific literature about the application of MSCs and MSC-EVs in the treatment of various infection diseases. The paper il well written, clear and interesting. I have only minor comments.

1)Table 1: please insert the references that report the markers expressed by the MSCs

2)Line 17. The authors reported that "pre-treated MSCs demonstrated  enhanced differentiation efficiency". Please specify the treatment they are referring to.

Author Response

Reviewer 2

COMMENT 1: Table 1: please insert the references that report the markers expressed by the MSCs

ANSWER 1: References were inserted into Table 1.

COMMENT 2: Line 17. The authors reported that “pre-treated MSCs demonstrated enhanced differentiation efficiency”. Please specify the treatment they are referring to.

ANSWER 2: The MSC transplantation increases osteoblast differentiation, blocks osteoclast activation, and rebalances bone formation and resorption. MSC transplantation in the treatment of osteoporosis enhances osteogenic differentiation, increases bone mineral density and prevents the progression of osteoporosis. In addition, gene modification of MSCs with important osteogenic genes is a promising approach to enhance their therapeutic effects. Besides, Sox11 is an important regulator of MSC differentiation and migration. It has been shown that Sox11-modified MSCs is promising for accelerating bone fracture healing, which may reduce delayed unions or non-unions [115, Xu, et al, 2015].

Reviewer 3 Report

Authors reviewed basic research to clinical practice using mesenchymal stem cells (MSCs) and MSCs-derived extracellular vesicles (MSC-EVs) in the infectious diseases. The review article seems to be timely during the COVID-19 pandemic periods for clinicians and researchers.

1.        For better understanding of MSCs and MSC-EVs applied for the infectious diseases, it is necessary to provide graphic Figures (instead of Figures 2 and 3) explaining profound mechanisms underlying an immunomodulatory and anti-inflammatory interactions between MSCs/MSC-EVs and immune cells, as well as antiviral/antibacterial effects with promotion of the restoration of the epithelium and tissue regeneration. It is better to accumulate several sentences seen in the lines 207–236 and liens 386–392.

2.        Figure 1 needs to be replaced as Supplement files because of unsuitable presentation as a scientific report.

3.        Functional differences between MSCs and MSC-EVs need to be clearly stated, especially superiority of MSC-EVs for clinical application in 2. MSCs and MSC-derived extracellular vesicles (MSC-EVs). 

4.        Authors need to add Tables or supplement Tables reviewing clinical trials of the infectious diseases in 3. Viral infectious diseases and 4. Bacterial infectious diseases. 

5.        The quality of the paragraphs seems to be different, therefore authors need to revise the text reviewed by a native English speaker. 

6.        Is the time correct in lines 323–324? Authors need to revise the manuscript as correct as possible.

Author Response

Reviewer 3

COMMENT 1: For better understanding of MSCs and MSC-EVs applied for the infectious diseases, it is necessary to provide graphic Figures (instead of Figures 2 and 3) explaining profound mechanisms underlying an immunomodulatory and anti-inflammatory interactions between MSCs/MSC-EVs and immune cells, as well as antiviral/antibacterial effects with promotion of the restoration of the epithelium and tissue regeneration. It is better to accumulate several sentences seen in the lines 207–236 and lines 386–392.

ANSWER 1: Tables 2. Mechanisms of MSCs and MSC-EVs influence on immune cells” was added. Figures 2 and 3 were corrected.

COMMENT 2: Figure 1 needs to be replaced as Supplement files because of unsuitable presentation as a scientific report.

ANSWER 2: Figure 1 was replaced as Supplement 1.  

COMMENT 3: Functional differences between MSCs and MSC-EVs need to be clearly stated, especially superiority of MSC-EVs for clinical application in 2. MSCs and MSC-derived extracellular vesicles (MSC-EVs). 

ANSWER 3: MSCs have been found to play an immunomodulatory role in numerous infection diseases through the production of soluble factors, and the transfer of EVs containing various molecules [68, Mardpour, et al., 2019]. It has been established that MSC-EVs have the same immunomodulatory and anti-inflammatory and other effects as their parental cells and recapitulate a broad range of the therapeutic effects shown by MSC treatment [69, Kim, et al., 2020]. However, there are different mechanisms underlying the interaction of various MSCs or MSC-EVs with immune cells. EVs derived from different types of MSCs have similar and unique characteristics (Table 2). 

Currently, most clinical trials of MSC therapy for viral and bacterial infection diseases have focused on patients who have not responded to traditional disease drug therapy as COVID-19, AIDS, and tuberculosis. However, the use of MSCs in therapeutic treatments still has many challenges. An increasing number of studies reveal that MSCs are highly heterogeneous with different multipotential properties, progenitors, and cell states. In addition, MSCs isolated from different sources exhibit distinct characteristics, known as tissue sources-associated heterogeneity

[86, Costa, e al., 2021; 87, Levy et al., 2020; 88, Brown et al., 2019].

Besides, the intravenous administration of MSCs can lead to aggregation or clumping of cells in the vascular system and associated with the risk of mutagenicity and oncogenicity [89, Tavakoli, et al., 2020; 90, Tang, et al., 2021].

The application of MSC-EVs in the treatment of diseases is a novel concept with particular advantages over the whole-cell therapy. MSC-EVs are well tolerated and have low immunogenicity and also have a more stable membrane structure than MSCs. Another advantage of MSCs-EVs over MSCs is the possibility of storing them for several weeks/months allowing their safe transportation and delayed therapeutic use [96, Lai, et al., 2019].

These advantages of EVs provide broader prospects for disease treatment. However, the studying of the mechanism of EVs in the treatment of diseases is the primary connection to future clinical research.

COMMENT 4: Authors need to add Tables or supplement Tables reviewing clinical trials of the infectious diseases in 3. Viral infectious diseases and 4. Bacterial infectious diseases.

ANSWER 4: Tables 3 and 4 were added. Currently, studies are predominantly carried out to evaluate the clinical efficacy of MSCs and MSC-EVs in vitro and in vivo, there are only a few clinical trials on this topic (Table 4).

COMMENT 5: The quality of the paragraphs seems to be different, therefore authors need to revise the text reviewed by a native English speaker.

ANSWER 5: We have revied the manuscript accordingly.

COMMENT 6: Is the time correct in lines 323–324? Authors need to revise the manuscript as correct as possible.

ANSWER 6: We have revied the manuscript accordingly.

Round 2

Reviewer 3 Report

Authors comprehensively revised the manuscript answering reviewers comments. Figures are fine. It is an interesting review article for readers.

Minor spelling errors need to be corrected including in the Tables, for example, alpha, beta, and gamma as well as the same abbreviation several times found in the text.

Author Response

Minor spelling errors were be corrected.